# Done Is Better than Perfect: Unlocking Efficient Reasoning by Structured Multi-Turn Decomposition

## Abstract

Large Reasoning Models (LRMs) have gained increasing attention over the past few months. Despite being effective, LRMs are criticized for the excessively lengthy Chain-of-Thought (CoT) to derive the final answer, suffering from high first-token and overall latency. Typically, the CoT of LRMs mixes multiple *thinking units*, some of which are split by markers like "aha", "wait", or "alternatively"; each unit attempts to produce a candidate answer to the original query. Hence, a natural idea to improve efficiency is to reduce the unit number. Yet, the fact that the thinking units in vanilla CoT cannot be explicitly managed renders doing so challenging. This paper introduces **Multi-Turn Decomposition (MinD)** to decode conventional CoT into a sequence of explicit, structured, and turn-wise interactions to bridge the gap. In MinD, the model provides a multi-turn response to the query, where each turn embraces a thinking unit and yields a corresponding answer. The subsequent turns can reflect, verify, revise, or explore alternative approaches to both the thinking and answer parts of earlier ones. This not only makes the answer delivered more swiftly, but also enables explicit controls over the iterative reasoning process (i.e., users may halt or continue at any turn). We follow a supervised fine-tuning (SFT) then reinforcement learning (RL) paradigm to realize MinD. We first rephrase the outputs of an LRM into multi-turn formats by prompting another LLM, and then tune the LRM with such data. Observing that the tuned model tends to consume even more tokens than the original one (probably due to that the multi-turn formats introduce additional answer tokens), we advocate leveraging RL algorithms like GRPO to prioritize correct outputs with fewer turns. Trained on the MATH dataset using R1-Distill models, MinD can achieve up to $\sim 70\%$ reduction in both output token usage and time to first token (TTFT), while maintaining competitive performance on benchmarks such as MATH-500, AIME24, AMC23, GPQA-Diamond, and LiveCodeBench.

## 1 Introduction

Large Reasoning Models (LRMs) have recently attracted significant attention due to their advancing reasoning capabilities, including OpenAI-o1 (Jaech et al., 2024), DeepSeek-R1 (Guo et al., 2025), and Kimi-1.5 (Kimi et al., 2025). These models have achieved remarkable performance on complex tasks, e.g., mathematical competitions, thanks to their ability to engage in a "think-then-answer" paradigm, where intermediate reasoning chains are generated to induce the final answer. The resultant Chain-of-Thought (CoT) activates contextually accurate responses through iterative exploration and verification of potential solutions.

Despite these advantages, LRMs often suffer from inefficiency issues as the CoT can become excessively lengthy, exhibiting substantially increased computational costs and latency compared to non-reasoning Large Language Models (LLMs). To mitigate these, several strategies have been proposed in recent works. For example, some approaches encourage models to generate answers more directly through strategically designed prompts (Jie et al., 2024), truncate the chain of thought to avoid unnecessary token generation (Fu et al., 2025; Qwen, 2025), or leverage speculative reasoning via model collaboration (Pan et al., 2025; She et al., 2025). Other approaches focus on reducing token redundancy by refining model reasoning paths through supervised fine-tuning (SFT) (Yang

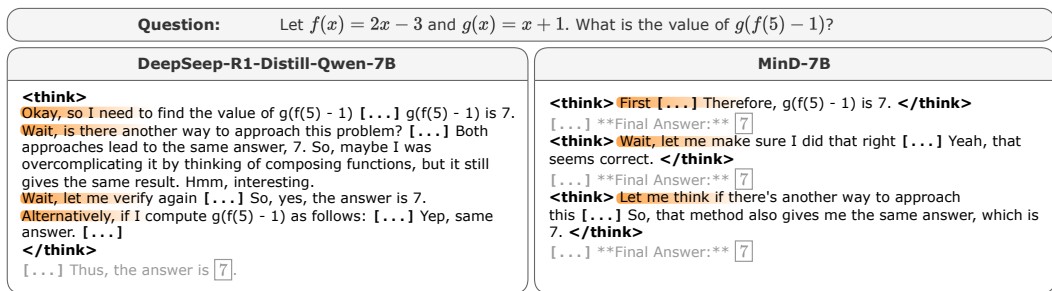

Figure 1: An illustration of responses from DeepSeek-R1-Distill-Qwen-7B and the transformed MinD-7B model on the same math problem. The original LRM follows a think-then-answer format, where the reasoning process consists of multiple thinking units (the start of each new unit is marked with an orange highlight). In contrast, MinD-7B adopts a multi-turn reasoning paradigm, where each turn contains a thinking unit followed by an answer. Also note that MinD-7B tends to use fewer thinking units due to the GRPO training (see Section 3.3).

et al., 2025c), or by enhancing decision efficiency with improvements to Group Relative Policy Optimization (GRPO) algorithms (Yu et al., 2025; Liu et al., 2025).

The CoT reasoning process in LRMs is typically composed of multiple *thinking units*—discrete cognitive steps like initial attempts, follow-up validations, reflections, and strategic shifts. Each unit can contribute to generating a candidate answer, while current LRMs tend to employ redundant units to ensure the final answer is close to "perfect" (see an empirical analysis of such redundancy in Figure 2 (right)). While reducing the number of thinking units could improve reasoning efficiency, the inability to explicitly manage these units in standard CoT makes this challenging. This highlights the need for more fine-grained approaches to improve reasoning efficiency.

Building on this insight, we introduce **Multi-Turn Decomposition (MinD)** to decode the "think-then-answer" CoT reasoning into a sequence of multi-turn interactions to enable the explicit control of the number of thinking units, where each turn contains a single thinking unit and an answer generated based on both the current and all preceding units. Refer to Figure 1 for an illustration of the paradigm shift. To implement MinD, we adopt a pipeline combining SFT and GRPO. We first convert conventional CoT traces into structured, multi-turn formats using GPT-4o (OpenAI et al., 2024) and then fine-tune the target model on such data. To further enhance efficiency, we apply GRPO to encourage the model to generate accurate responses within fewer reasoning turns, thereby reducing latency and computational costs.

To evaluate the effectiveness of MinD, we conduct extensive experiments across a range of reasoning benchmarks. On DeepSeek-R1-Distill-Qwen-1.5B, MinD reduces token usage by up to $\sim 70\%$ and accelerates time to first token (TTFT) by $4.2\times$ on MATH-500, while maintaining over 95% accuracy. Furthermore, MinD demonstrates strong out-of-distribution generalization on this model, with token reductions of 69% on AIME24 and 53% on GPQA-Diamond. These results highlight the efficiency and broad applicability of MinD in diverse reasoning scenarios.

## 2 RELATED WORK

**Efficient Reasoning Paradigms** Since CoT prompting (Wei et al., 2022), explicit multi-step traces have improved LLM reasoning (Guo et al., 2025) but often at the cost of long outputs, high token usage, and latency (Chiang & yi Lee, 2024). To address redundancy, recent work reduces intermediate tokens while preserving quality: token skipping (Xia et al., 2024) and length-harmonizing pruning (Luo et al., 2025a) report sizable savings with competitive accuracy (Fu et al., 2025). Orthogonally, latent/hidden-thinking methods (e.g., Token-Assorted Mixing (Su et al., 2025), Hidden Thinking (Shen et al., 2025)) move computation off the visible token stream, yielding multi-fold throughput gains (Hao et al., 2025). Hybrid systems (e.g., C3OT (Kang et al., 2025)) and speculative pipelines (Pan et al., 2025; Zhang et al., 2024; She et al., 2025) further balance accuracy and compute via verification and adaptive depth.

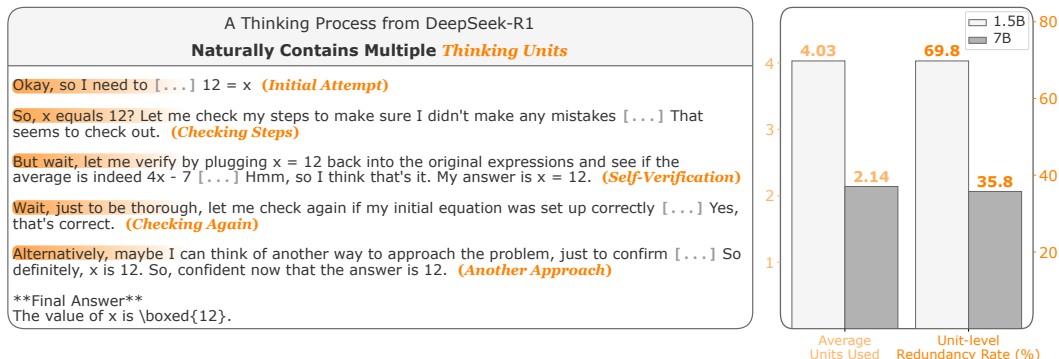

Figure 2: **Left:** An example of a standard CoT from DeepSeek-R1, naturally containing multiple discrete thinking units (the start of each new unit is marked with an orange highlight). **Right:** Empirical analysis of unit-level redundancy, which is calculated based on Equation (5), in R1-distilled models on the MATH-500 dataset, showing an average redundancy rate of 69.8% for the 1.5B model and 35.8% for the 7B model.

**Reinforcement Learning for Reasoning Optimization** Reinforcement learning (RL) has become an essential tool for optimizing LLM reasoning, providing precise control over decision-making processes. Group Relative Policy Optimization (GRPO) (Shao et al., 2024) is one of the most influential methods in this domain, aligning reward signals with step-wise reasoning validity rather than simply final answer correctness. This strategy allows models to prioritize accurate intermediate steps, enhancing both response precision and computational efficiency. Building on this foundation, frameworks like DAPO (Yu et al., 2025) and R1-Zero (Liu et al., 2025) incorporate dynamic reward shaping and entropy-controlled exploration to further refine model outputs. These methods extend GRPO by introducing adaptive mechanisms that reduce token redundancy while maintaining high accuracy, making them particularly effective for complex reasoning tasks. Recent advancements have also focused on integrating search-based techniques to enhance reasoning efficiency. For instance, Search-R1 (Jin et al., 2025) combines Monte Carlo Tree Search with policy gradients to optimize reasoning path selection, reducing unnecessary token usage. Similarly, length-aware control frameworks like L1-Controller (Aggarwal & Welleck, 2025) balance correctness and token efficiency through dual reward signals, achieving substantial latency reductions. Other approaches, such as R1-Searcher (Song et al., 2025), incorporate dynamic halting mechanisms to automatically terminate unproductive reasoning chains, significantly improving efficiency in open-domain tasks. ThinkPrune (Hou et al., 2025) adopts length clipping to the reward function, pruning outputs to reduce redundancy. ShorterBetter (Yi et al., 2025) uses the "Sample Optimal Length"—the shortest correct response as a self-supervised reward to guide models toward generating more concise traces without compromising accuracy. AdaptThink (Zhang et al., 2025) empowers models to adaptively choose thinking mode via a constrained optimization objective and importance sampling. SCoRe (Kumar et al., 2024) trains models via multi-turn RL to self-diagnose and correct errors from self-generated traces, prioritizing correctness over efficiency.

**Training-Based Efficiency Enhancements** Training strategies have also played a critical role in improving reasoning efficiency. Supervised fine-tuning (SFT) methods like Thinking-Optimal Scaling (Yang et al., 2025c) align models with optimal solution trajectories, reducing token redundancy without compromising accuracy. This approach effectively reshapes the internal reasoning paths of models, ensuring more concise outputs. Hybrid training regimes have also gained traction, combining imitation learning and reinforcement learning to refine reasoning efficiency. For example, the SpecReason framework (Pan et al., 2025) employs a two-stage process, beginning with teacher-student distillation for foundational policy approximation, followed by adversarial reward shaping for fine-grained optimization. This blend of supervised and reinforcement learning techniques has proven effective in reducing token counts while maintaining response quality.

# 3 METHOD

In this section, we first introduce the standard Chain-of-Thought (CoT) reasoning of Large reasoning models (LRMs) and briefly review Group Relative Policy Optimization (GRPO) (DeepSeek-AI, 2025). We then present an empirical study showing how redundant reasoning steps commonly arise in LRMs. Finally, we outline MinD, which reformulates the standard CoT into a multi-turn structure, and discuss how to leverage GRPO to encourage concise and effective multi-turn reasoning.

## 3.1 PRELIMINARY

**CoT for LRMs** LRMs commonly adopt a "think-then-answer" paradigm for complex problem solving. Given a query $q$, an LRM typically produces an output $o$ of the form:

$$q \rightarrow o = \texttt{<think>}\, t \,\texttt{</think>}\, a \,, \tag{1}$$

where $t$ denotes the internal thinking process, delimited by `<think>` and `</think>`, and $a$ is the final answer. The thinking process $t$ can be viewed as an exploration of the solution space and is naturally decomposed into multiple *thinking units*—self-contained logical steps that can induce a candidate answer to $q$, with an example from DeepSeek-R1 (Guo et al., 2025) depicted in Figure 2 (left). Formally, letting $u_i$ denote a thinking unit, there is $t = (u_1, u_2, \ldots, u_n)$. These units may arise from (1) an initial attempt to solve the problem, (2) depth-wise exploration such as validation, backtracking, or correction along a single line of reasoning, or (3) breadth-wise search involving alternative methods or perspectives. Each unit can thus be interpreted as a path in the reasoning space, potentially building on previous steps, and may terminate with a provisional answer to the query.

However, current LRMs tend to employ numerous thinking units before gaining the final answer to solve the problem as 'perfectly' as possible, causing significant inefficiency issues.

**GRPO** Let $\pi_\theta$ denote the current policy and $\pi_{\theta_{\text{old}}}$ the reference policy from the previous iteration. Given a query $q$, GRPO samples $G$ completions $o_1, \ldots, o_G$ and optimizes the objective:

$$\mathbb{E}_{q, \{o_i\}_{i=1}^G} \left[ \frac{1}{G} \sum_{i=1}^{G} \frac{1}{|o_i|} \sum_{j=1}^{|o_i|} \min\left( \rho_{i,j} A_i, \ \text{clip}(\rho_{i,j}, 1-\epsilon, 1+\epsilon) A_i \right) \right] \,, \tag{2}$$

where $\rho_{i,j} = \frac{\pi_\theta(o_{i,j}|q, o_{i,<j})}{\pi_{\theta_{\text{old}}}(o_{i,j}|q, o_{i,<j})}$ is the ratio between the new and old policies for token $j$ in sequence $o_i$ and $|o_i|$ is the sequence length. $A_i$ is the group-standardized advantage:

$$A_i = \frac{R(o_i) - \text{mean}(\{R(o_1), \ldots, R(o_G)\})}{\text{std}(\{R(o_1), \ldots, R(o_G)\})} \,, \tag{3}$$

where $R$ denotes the reward function, and $\text{mean}(\{r_1, \ldots, r_G\})$ and $\text{std}(\{r_1, \ldots, r_G\})$ represent the mean and standard deviation of group rewards, respectively. For clarity, we omit the KL regularization term, as it is not the focus of our analysis.

## 3.2 UNIT-LEVEL REDUNDANCY IN LRMS

Before devoting to reducing the number of thinking units of LRMs, we first systematically investigate the *unit-level redundancy*, which is intuitively high considering the repeated depth-wise validations or breadth-wise explorations of alternative solution paths, even after repeatedly arriving at essentially the same valid answer, in long CoTs.

Concretley, we conducted a detailed analysis using DeepSeek-R1-Distill-Qwen-1.5B/7B (DeepSeek-AI, 2025). We extracted their CoT traces from the MATH (Lightman et al., 2023) and GSM8K (Cobbe et al., 2021) training sets (restricted to correctly answered examples), and segmented each trace into discrete thinking units using GPT-4o (OpenAI et al., 2024) (see Appendix C for details).

For each segmented trace $t = (u_1, u_2, \ldots, u_n)$, we constructed prefix sub-traces $t_{\leq k} = (u_1, \ldots, u_k)$ for $1 \leq k \leq n$. We then prompted the model to generate an intermediate answer $a_k$ by appending a special stop token `</think>` after $t_{\leq k}$ given the current partial reasoning:

$$q \rightarrow o_k = \texttt{<think>}\, t_{\leq k} \,\texttt{</think>}\, a_k \,, \quad k = 1, \cdots, n \,. \tag{4}$$

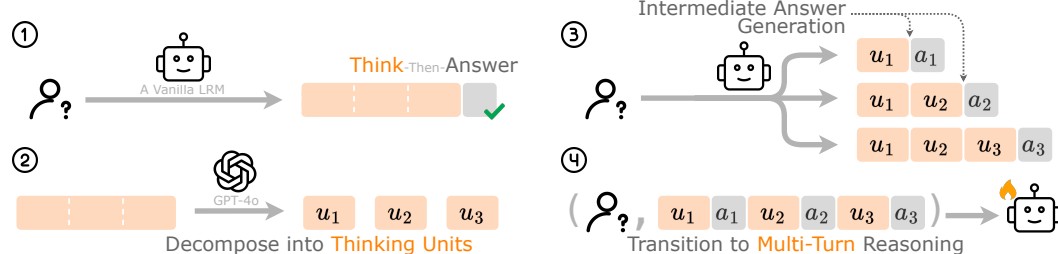

Figure 3: Transforming think-then-answer LRMs into a multi-turn reasoning paradigm, consisting of four steps: (1) Rejection sampling to filter out responses with correct final answers; (2) Unit segmentation using GPT-4o to divide CoTs into discrete reasoning units; (3) Intermediate answer completion to extract answers ($a_k$) for each prefix sub-trace ($t_{\leq k}$); and (4) SFT to align LRMs with the multi-turn format.

To quantify unit-level redundancy, we define the minimal sufficient prefix $t_{\leq n^*}$ as the shortest prefix that leads to a correct final answer. The *unit-level redundancy rate* is then defined as:

$$\text{URR} = \frac{n - n^*}{n} \cdot \mathbb{1}_{a_n \text{ is correct}} , \qquad (5)$$

where $n$ is the total number of thinking units and $n^*$ is the minimal number required for correctness. A higher URR indicates a greater proportion of unnecessary reasoning steps.

Our empirical results, summarized in Figure 2 (right), show that the average unit-level redundancy rates are 69.8% for the 1.5B model and 35.8% for the 7B model. This reveals that a significant portion of the reasoning process in current LRMs is redundant for solving the problem, underscoring the potential for substantial efficiency gains by explicitly mitigating unit-level redundancy.

### 3.3 MULTI-TURN DECOMPOSITION (MIND)

Our basic notion is that the model should not be that cautious. Given that "done is better than perfect", we aim to let the model yield a candidate answer as soon as possible. Besides, we would also like to penalize the unit-level redundancy. MinD realizes these through two key innovations.

**Multi-Turn CoT Reformulation**  MinD first employs supervised fine-tuning (SFT) to shift the reasoning paradigm from "think-then-answer" (i.e., Equation (1)) to a structured multi-turn format:

$$\texttt{<think>}\, u_1 \,\texttt{</think>}\, a_1 \,\texttt{<think>}\, u_2 \,\texttt{</think>}\, a_2 \cdots \texttt{<think>}\, u_n \,\texttt{</think>}\, a_n , \quad (6)$$

where the thinking units $(u_1, u_2, \ldots, u_n)$ in the original CoT $t$ are distributed into a sequence of *reasoning turns*. Each turn also includes an intermediate answer $a_k$.

To construct the training data for multi-turn SFT, we first segment the original thinking process $t$ into $(u_1, u_2, \ldots, u_n)$, and then generate an intermediate answer $a_k$ after each $u_k$, as described in Section 3.2. The overall pipeline is illustrated in Figure 3.

After training, the learned multi-turn LRM enables flexible management of the thinking units (e.g., an external controller can choose to continue or abort from the reasoning by manipulating the token `</think>`), but we empirically observe that when applying no control, the model tends to generate even more output tokens than the original one (see Table 4). This is because SFT primarily reshapes the reasoning format without directly addressing unit-level redundancy, and $a_k$ incurs further token usage. To bridge the gap, we suggest leveraging GRPO to prioritize efficient reasoning traces.

**Reducing Reasoning Turns via GRPO**  We define a reward function $R$ comprises three components for GRPO:

$$R = \mathcal{R}_{\text{format}} + \mathcal{R}_{\text{accuracy}} + \mathcal{R}_{\text{unit}} . \qquad (7)$$

In detail, they are: (1) Format Consistency Reward $\mathcal{R}_{\text{format}}$, which ensures that the generated output adheres to the multi-turn structure described in Equation (6). (2) Answer Accuracy Reward $\mathcal{R}_{\text{accuracy}}$,

which rewards the model for producing a correct final answer, as determined by matching $a_n$ to the ground truth. (3) Unit Compactness Reward $\mathcal{R}_{\text{unit}}$, which penalizes cases where a single reasoning unit contains multiple exploratory trajectories and thus encourages a clear separation between reasoning turns. Concretely, we treat a unit as "overloaded" when it contains linguistic cues that typically signal a restart or alternative line of thought (e.g., phrases like "double-check", "wait", or "alternatively" appearing multiple times within the same unit). This detection is implemented as a simple pattern-based heuristic over the generated text, without invoking any external LLM, and therefore adds negligible cost beyond standard GRPO. The specific weights for each reward component are detailed in Table 1, and we analyze the empirical effect of $\mathcal{R}_{\text{unit}}$ in Section 4.3.

Note that we do not introduce an explicit reward term regarding the number of turns, because GRPO inherently introduces an implicit bias toward generating shorter CoTs that yield correct answers. As shown in Equation (2), for a fixed advantage $A_i$, the per-token normalization $1/|o_i|$ results in larger per-token updates for shorter outputs (Lin et al., 2025; Yu et al., 2025; Liu et al., 2025), thereby encouraging the model to produce more concise and efficient completions. This effect is particularly pronounced in LRMs, which typically possess strong reasoning capabilities and can generate multiple correct yet diverse completions per group during training. Thus, the GRPO framework naturally incentivizes the model to favor responses with fewer reasoning turns. This behavior is empirically validated in Figure 5, where we observe a substantial reduction in the number of reasoning turns following GRPO training.

## 4    EXPERIMENTS

In this section, we evaluate the efficiency of MinD across several benchmarks. Section 4.1 describes the experimental setup. More detailed settings can be found in Appendix B. Section 4.2 presents the main results, focusing on token reduction, accuracy, and latency. Ablation studies and additional discussion are provided in Section 4.3.

### 4.1    SETUP

Table 1: Reward function value settings.

|  | $\mathcal{R}_{\text{format}}$ | $\mathcal{R}_{\text{accuracy}}$ | $\mathcal{R}_{\text{unit}}$ |
|---|---|---|---|
| Compliance | +1 | +2 | 0 |
| Non-Compliance | -1 | -2 | -0.3 |

Table 2: Training data sizes.

|  | 1.5B | 7B |
|---|---|---|
| SFT | 3610 | 3532 |
| GRPO | 7500 | 7500 |

**Training Details**    The training process for MinD consists of two key phases, as described in Section 3.3. The first SFT phase is conducted using the LLaMA-Factory repository (Zheng et al., 2024). We perform full-parameter fine-tuning for 2 epochs with a learning rate of 5e-5. The second GRPO phase leverages the veRL repository (Sheng et al., 2024). During this phase, we train for 1 epoch with an actor learning rate of 1e-6. For each training step, 10 roll-out completions are generated for each sample, with all other hyperparameters set to the default values provided by veRL. The reward function described in Section 3.3 is adopted with the weight configurations listed in Table 1.

**Models & Datasets**    We conduct our experiments using DeepSeek-R1-Distill-Qwen-1.5B/7B (DeepSeek-AI, 2025). For SFT, the training data consists of questions from the GSM8K (Cobbe et al., 2021) and MATH (Lightman et al., 2023) training sets. Model-generated responses are filtered via rejection sampling to retain only correct answers, then pre-processed as shown in Figure 3. For GRPO, we use the MATH training set exclusively, with sample sizes detailed in Table 2. We evaluate on both in-distribution (MATH-500 (Lightman et al., 2023)) and out-of-distribution benchmarks, including AIME24 (of America, 2024), AMC23 (of Science, 2023), GPQA-Diamond (Rein et al., 2023), and LiveCodeBench (24.10–25.01) (Jain et al., 2024), to assess generalization. Additional results on more models and benchmarks are provided in Tables 5 and 6.

**Baselines**    To assess the efficiency of our method, we compare against the following baselines: Original LRM: The base models used in this work, DeepSeek-R1-Distill-Qwen-1.5B and 7B.

Table 3: Performance comparison of various baselines and our proposed method, MinD, across five reasoning benchmarks: MATH-500, AIME24, AMC23, GPQA-Diamond, and LiveCodeBench (2024.10–2025.01). We report accuracy (Acc.; higher is better) and average output token usage (Tokens; lower is better) for both 1.5B and 7B configurations. Methods include the original LRM (DeepSeek-R1-Distill-Qwen-1.5B/7B), ThinkPrune (Hou et al., 2025), Dynasor (Fu et al., 2025), DEER (Yang et al., 2025b), ShorterBetter (Yi et al., 2025), AdaptThink (Zhang et al., 2025), and our method, MinD. **MinD is trained only on the MATH training set**, making MATH-500 in-domain and the other benchmarks out-of-domain. As shown, MinD delivers competitive or superior accuracy while substantially reducing token usage, demonstrating efficient and generalizable reasoning. Some entries are omitted because the original papers did not report the corresponding results and reliable reproduction was not feasible.

| | MATH-500 | | AIME24 | | AMC23 | | GPQA-Diamond | | LiveCodeBench | |
|---|---|---|---|---|---|---|---|---|---|---|
| | Acc.↑ | Tokens↓ | Acc.↑ | Tokens↓ | Acc.↑ | Tokens↓ | Acc.↑ | Tokens↓ | Acc.↑ | Tokens↓ |
| **1.5B** | | | | | | | | | | |
| Original LRM | 85.4 | 5389 | 26.7 | 15177 | 67.5 | 9956 | 32.3 | 9842 | 12.0 | 21960 |
| ThinkPrune | $81.6_{-4.4\%}$ | $2427_{-55\%}$ | $31.6_{+18.4\%}$ | $7700_{-49\%}$ | $69.2_{+2.5\%}$ | $4074_{-59\%}$ | $31.3_{-3\%}$ | $6474_{-34\%}$ | $10.2_{-15\%}$ | $18463_{-16\%}$ |
| DEER | $73.2_{-14.3\%}$ | $1118_{-79\%}$ | $20.0_{-25.1\%}$ | $3302_{-78\%}$ | $47.5_{-29.6\%}$ | $2384_{-76\%}$ | $5.6_{-82.7\%}$ | $4128_{-58\%}$ | - | - |
| ShorterBetter | $74.8_{-12.4\%}$ | $1008_{-81\%}$ | $21.3_{-20.2\%}$ | $3705_{-76\%}$ | $65.3_{-3.3\%}$ | $2206_{-78\%}$ | $33.3_{+3.1\%}$ | $4362_{-56\%}$ | $11.6_{-3.3\%}$ | $9284_{-58\%}$ |
| AdaptThink | $82.0_{-4.0\%}$ | $1884_{-65\%}$ | $29.0_{+8.6\%}$ | $7171_{-53\%}$ | $71.3_{+5.6\%}$ | $3706_{-63\%}$ | $35.8_{+10.8\%}$ | $8083_{-18\%}$ | $12.3_{+2.5\%}$ | $15240_{-31\%}$ |
| MinD | $82.8_{-3.0\%}$ | $1719_{-68\%}$ | $30.8_{+15.4\%}$ | $4676_{-69\%}$ | $75.6_{+12.0\%}$ | $2432_{-76\%}$ | $31.3_{-3.1\%}$ | $4690_{-52\%}$ | $12.7_{+5.8\%}$ | $17728_{-19\%}$ |
| **7B** | | | | | | | | | | |
| Original LRM | 93.0 | 3928 | 50.0 | 14107 | 90.0 | 6076 | 50.5 | 8390 | 34.3 | 13690 |
| Dynasor | $88.5_{-4.8\%}$ | $2591_{-34\%}$ | $47.7_{-4.6\%}$ | $8760_{-38\%}$ | $87.1_{-3.2\%}$ | $4913_{-19\%}$ | - | - | - | - |
| DEER | $90.2_{-3.0\%}$ | $2391_{-39\%}$ | $49.2_{-1.6\%}$ | $10046_{-29\%}$ | $87.5_{-2.8\%}$ | $4877_{-20\%}$ | $30.6_{-39.4\%}$ | $5682_{-32\%}$ | - | - |
| ShorterBetter | $90.0_{-3.2\%}$ | $1272_{-67.6\%}$ | $53.3_{+6.6\%}$ | $5288_{-63\%}$ | $83.6_{-7.1\%}$ | $1946_{-68\%}$ | $49.6_{-1.7\%}$ | $4257_{-49\%}$ | $30.1_{-12.2\%}$ | $9067_{-34\%}$ |
| AdaptThink | $91.8_{-1.3\%}$ | $2547_{-35\%}$ | $55.1_{+10.2\%}$ | $8623_{-39\%}$ | $90.3_{+0.3\%}$ | $3457_{-43\%}$ | $50.3_{-0.5\%}$ | $7527_{-10\%}$ | $31.4_{-8.5\%}$ | $9586_{-30\%}$ |
| MinD | $91.6_{-1.5\%}$ | $2859_{-27\%}$ | $45.4_{-9.2\%}$ | $7588_{-46\%}$ | $92.0_{+2.2\%}$ | $3729_{-39\%}$ | $53.0_{+5.0\%}$ | $6845_{-18\%}$ | $34.0_{-0.9\%}$ | $10113_{-26\%}$ |

ThinkPrune (Hou et al., 2025): Adds length clipping to the GRPO reward and is trained on the AIME-AMC subset, progressively pruning outputs at the token level to reduce response length. DEER (Yang et al., 2025b): A training-free approach that detects "action transition points" (e.g., "Wait," "Alternatively," "Hmm") to trigger answer generation, and halts decoding when the mean token probability surpasses a confidence threshold. Dynasor (Fu et al., 2025): Periodically inserts probes (e.g., every 32, 64, or 128 tokens) to extract intermediate answers and assess their consistency, enabling early termination of generation. ShorterBetter (Yi et al., 2025): Determines the shortest correct CoT across multiple samples as a dynamic reward to guide models toward generating more concise traces. AdaptThink (Zhang et al., 2025): An RL-based post-training method that combines a constrained objective with importance-sampled training to empower models to adaptively choose between thinking and non-thinking modes. Both ShorterBetter and AdaptThink are trained on the DeepScaleR (Luo et al., 2025b).

**Evaluation Metrics** We evaluate MinD using three primary metrics: accuracy, average output token usage, and time-to-first-token (TTFT). TTFT measures the time it takes for the model to generate the first answer token of the response, from when the prompt was sent—a key determinant of user experience. The evaluations are conducted using the Open-R1 evaluation scripts (Face, 2025), with a maximum sequence length of 32,768 tokens, a temperature setting of 0.6, and a top-p value of 0.95, running on four NVIDIA A100 GPUs.

## 4.2 MAIN RESULTS

**Reducing Output Tokens for Efficient Reasoning** After training the 1.5B and 7B multi-turn reasoning models as described in Section 4.1, we evaluated their token efficiency across a range of reasoning benchmarks. The results, summarized in Table 3, show that MinD consistently reduces output token usage while maintaining strong performance. On in-domain MATH-500, MinD lowers

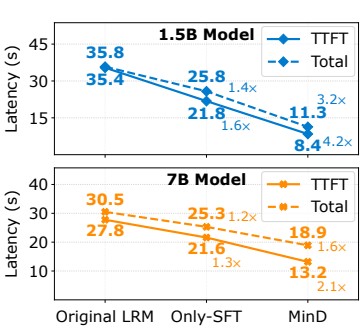 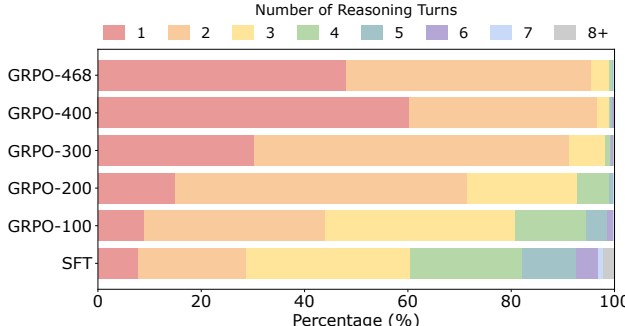

Figure 4: TTFT (time to first token) and total latency of two DeepSeek-R1-distilled models on MATH-500. MinD achieves up to 4.2× (1.5B) and 2.1× (7B) speedups over the original LRMs in TTFT, and 3.2× (1.5B) and 1.6× (7B) in total latency.

Figure 5: The distribution of reasoning turns for MinD at different training stages (1.5B model) on the MATH-500 dataset. Each bar represents a model checkpoint, including the SFT model and successive GRPO training steps. As GRPO training progresses, the number of reasoning turns per output decreases and becomes increasingly concentrated at 1 or 2 turns (highlighted in red and orange), demonstrating the effectiveness of GRPO in mitigating reasoning redundancy.

the average token usage to 1719 for the 1.5B model—a 68% reduction from the Original LRM (5389 tokens)—while achieving 82.8% accuracy. Although ThinkPrune attains similar accuracy (83.2%), it requires more tokens (1938). DEER achieves the lowest token usage (1118), but with a substantial accuracy drop to 73.2%. For the 7B model, MinD reduces average token usage by 27% compared to the Original LRM (2859 vs. 3928), with a high accuracy of 91.6%, outperforming both Dynasor and DEER in the balance of accuracy and efficiency. MinD's efficiency generalizes well to out-of-domain benchmarks. For example, on AMC23 (1.5B), MinD reaches 77.5% accuracy with 2384 tokens, substantially outperforming ThinkPrune and DEER in both accuracy and token reduction. Similar trends are observed on AIME24, GPQA-Diamond, and LiveCodeBench. These results demonstrate that MinD effectively eliminates unnecessary reasoning steps, producing concise, efficient outputs without compromising performance.

**Reducing TTFT and Total Latency** The TTFT and total response latency for the original R1-distilled LRMs and our MinD models are summarized in Figure 4. As shown, MinD significantly reduces both TTFT and total latency across both model sizes. For the 1.5B configuration, the original 1.5B model requires 35.4s TTFT, which drops to 21.8s after SFT and further to 8.4s with MinD, resulting in a 4.2× speedup. The total latency is similarly reduced from 35.8s (original) to 25.8s (SFT) and 11.3s (MinD), a 2.1× improvement. For the 7B model, TTFT decreases from 27.8s (original) to 21.6s (SFT) and 13.2s (MinD), achieving a 2.1× speedup. The total latency is reduced from 30.5s to 25.3s and 18.9s, for a 1.6× speedup. These results show that MinD shortens both the time to first answer token and the overall response latency, making the models more responsive.

## 4.3 DISCUSSION & ABLATION

**The Importance of Multi-Turn Structure** To evaluate the impact of the multi-turn design, we performed SFT using responses from the original distilled-1.5B model, without applying any multi-turn segmentation (i.e., using the same question set as in step (1) of Figure 3), followed by GRPO with only the format and outcome rewards. As shown in Table 4, the Non-Multi-Turn model achieves comparable results to MinD on in-distribution MATH-500, but exhibits a notable drop in accuracy and only marginal reductions in token usage on out-of-distribution benchmarks. We hypothesize that, under the conventional CoT format, models lack the flexibility to adjust the number of thinking units, making it difficult to learn a reasoning process that is both controllable and generalizable.

**GRPO is Crucial for Efficient Reasoning** As discussed in Section 3.3, SFT alone does not guarantee efficient reasoning. To demonstrate this, we compare the performance of models after SFT and after the full MinD pipeline, as shown in Table 4. The results reveal that SFT-only training

Table 4: Comparison of different training strategies on DeepSeek-R1-Distill-Qwen-1.5B. Original LRM refers to the pretrained baseline. SFT-Only applies only the supervised fine-tuning step from MinD. Non-Multi-Turn applies GRPO without explicit multi-turn segmentation. MinD denotes our full method with both multi-turn segmentation and GRPO. Acc.↑ indicates accuracy (higher is better), and Tokens↓ indicates average output length (lower is better).

| | Original LRM | | SFT-Only | | Non-Multi-Turn | | MinD | |
|---|---|---|---|---|---|---|---|---|
| | Acc.↑ | Tokens↓ | Acc.↑ | Tokens↓ | Acc.↑ | Tokens↓ | Acc.↑ | Tokens↓ |
| MATH-500 | 85.4 | 5389 | 82.8 | 5655 | 82.0 | 1866 | 82.8 | 1719 |
| AIME24 | 26.7 | 15177 | 26.7 | 20675 | 20.0 | 7654 | 30.8 | 4676 |
| AMC23 | 67.5 | 9956 | 77.5 | 8409 | 65.0 | 3415 | 75.6 | 2432 |
| GPQA-Diamond | 32.3 | 9842 | 28.3 | 12501 | 28.8 | 3397 | 31.3 | 4690 |

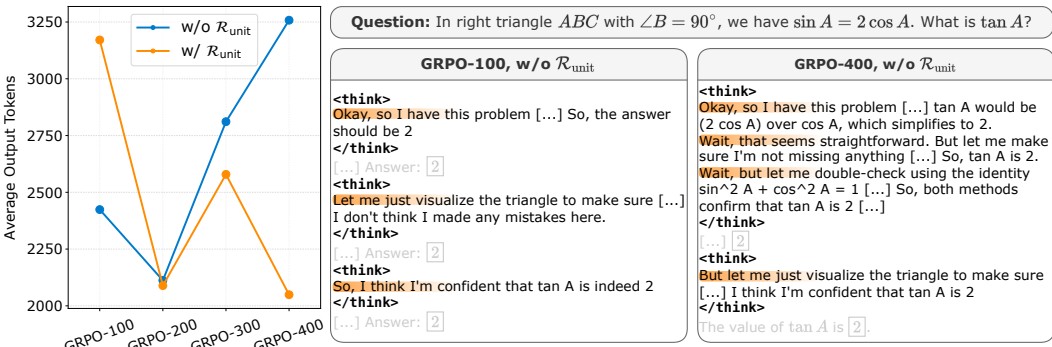

Figure 6: **Left:** Comparison of GRPO training with and without $\mathcal{R}_{\text{unit}}$ on MATH-500 for different 1.5B model checkpoints, showing Average Output Tokens for each. Removing $\mathcal{R}_{\text{unit}}$ leads to instability and collapse in output length. **Right:** An illustrative case comparing the outputs of GRPO-100-step and GRPO-400-step checkpoints trained without $\mathcal{R}_{\text{unit}}$. While the earlier checkpoint (GRPO-100) maintains clear multi-turn reasoning, the later checkpoint (GRPO-400) exhibits several thinking units within a single turn (the start of each new unit is marked with an orange highlight), demonstrating that omitting $\mathcal{R}_{\text{unit}}$ results in blurred step boundaries and loss of controllable, structured reasoning.

often increases average output token usage relative to the original LRM. In contrast, applying GRPO further leads to substantial reductions in token usage while preserving accuracy, underscoring the essential role of GRPO in enabling concise and effective reasoning.

**Role of $\mathcal{R}_{\text{unit}}$ in Maintaining Multi-Turn Reasoning** As discussed in Section 3.3 and detailed in Table 1, our GRPO framework introduces a Unit Compactness Reward, $\mathcal{R}_{\text{unit}}$, to enforce that each reasoning turn contains only a single, coherent exploratory trajectory. This mechanism is essential for preventing the model from degenerating into the original monolithic think-then-answer style—a common outcome under GRPO's token-level averaging (Section 3.3), which tends to favor shorter correct outputs. Without a specific penalty for multi-trajectory turns, the model may skip intermediate answers, collapsing the multi-turn reasoning structure into a single-block CoT. ~~To counteract this, $\mathcal{R}_{\text{unit}}$ penalizes reasoning turns that contain multiple exploratory trajectories, detected by linguistic cues such as phrases like "double-check."~~ This strategy encourages each turn to contain only one exploratory trajectory—especially in the critical first turn—without requiring external supervision, and thus maintains the multi-turn paradigm throughout training. The impact of $\mathcal{R}_{\text{unit}}$ is demonstrated in Figure 6, which shows how its absence leads to a collapse in output structure and length.

**MinD Effectively Alleviates Redundancy** To demonstrate the effectiveness of GRPO in reducing redundancy, we plotted the distribution of reasoning turns for SFT and GRPO models on the MATH-500 dataset, as shown in Figure 5. The figure clearly illustrates that GRPO significantly reduces the number of reasoning turns, indicating a more compact and efficient reasoning process compared to

the purely SFT-trained models. Additionally, from the data in Table 3, GRPO reduces the average output tokens on MATH-500 by 68.1% for the 1.5B model and 27.2% for the 7B model, compared to their respective original LRMs. This aligns well, though not directly, with the redundancy rates of 69.8% and 35.8% for these models, as reported in Figure 2 (Right). While these figures cannot be directly equated, they collectively indicate that MinD, through GRPO, substantially alleviates redundancy, resulting in more concise and efficient outputs.

Additional discussion can be found in Appendix A.

## 5  CONCLUSION

In this paper, we introduced Multi-Turn Decomposition (MinD), an efficient method for improving the reasoning efficiency of large language models. By structuring the reasoning process into multi-turn steps, MinD significantly reduces token usage and response latency while maintaining strong performance across various reasoning tasks. Our results demonstrate that structured reasoning provides a practical solution to challenges such as slow response times and high computational costs in large language models. A promising direction is adaptive multi-turn strategies that dynamically allocate reasoning turns according to task difficulty and user preferences.

## ETHICS STATEMENT

We acknowledge and adhere to the ICLR Code of Ethics for the entirety of this work. This study does not involve human subjects or sensitive personal data. All experiments use public benchmarks under their respective terms, with proper attribution. Our contribution aims to improve the accuracy–efficiency balance of reasoning models; nonetheless, deployment should follow standard safety safeguards (e.g., usage policies and filtering). No confidential or proprietary information was shared with third-party services. We disclose limited LLM assistance strictly for language editing, with human verification of all scientific content (Appendix D). The authors are solely responsible for the content of this paper.

## REPRODUCIBILITY STATEMENT

We aim to make all results reproducible. Model, training, and decoding details—including the MinD design, GRPO settings—are documented in the Method and Experiments sections; sensitivity analyses (e.g., the unit-compactness reward $\mathcal{R}_{\text{unit}}$) appear in Table 7. We will release a complete, reproducible codebase and configuration files upon acceptance.

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

## A MORE RESULTS

**Generalization across model families and tasks** To further test whether MinD is tied to a specific backbone or domain, we apply it to two additional LRMs on MATH-500: DeepSeek-R1-Distill-Llama3.1-8B (DeepSeek-AI, 2025) and Qwen3-1.7B (Yang et al., 2025a), summarized in Table 5. On R1-Distill-Llama3.1-8B, MinD reduces the average output length from 4792.37 to 3107.89 tokens (about 35% fewer tokens) while maintaining almost the same accuracy (77.4% vs. 78.0%). On Qwen3-1.7B, MinD achieves 89.2% accuracy compared to 91.0% for the original model, but uses only 3866.69 tokens on average instead of 5216.44 (around 26% reduction).

We further evaluate MinD on the non-mathematical OpenBookQA (Mihaylov et al., 2018) benchmark using DeepSeek-R1-Distill-1.5B (Table 6). In this setting, MinD improves accuracy from 27.4% to 34.8% while reducing token usage from 4986.47 to 3840.93 (about 23% fewer tokens). These results suggest that the unit-level multi-turn reformulation and RL training in MinD generalize across different model families and extend beyond purely mathematical reasoning tasks.

Table 5: Performance of MinD on additional LRMs on MATH-500.

| Model | Accuracy (%) | Tokens |
|---|---|---|
| R1-Llama3.1-8B | 78.0 | 4792.37 |
| R1-Llama3.1-8B-SFT | 75.2 | 5068.48 |
| R1-Llama3.1-8B-MinD | 77.4 | 3107.89 |
| Qwen3-1.7B | 91.0 | 5216.44 |
| Qwen3-1.7B-SFT | 88.6 | 5433.30 |
| Qwen3-1.7B-MinD | 89.2 | 3866.69 |

Table 6: Results of MinD on OpenBookQA with DeepSeek-R1-Distill-1.5B.

| Model | Accuracy | Tokens |
|---|---|---|
| R1-1.5B | 27.4 | 4986.47 |
| R1-1.5B-SFT | 31.0 | 5433.30 |
| R1-1.5B-MinD | 34.8 | 3840.93 |

**Ablation on the Unit-Compactness Reward Weight $\mathcal{R}_{\text{unit}}$** We study how the weight on the $\mathcal{R}_{\text{unit}}$ affects MinD's accuracy–efficiency trade-off. Specifically, we vary the non-compliance penalty for $\mathcal{R}$unit as specified in Table 1. Unless otherwise noted, all runs in this ablation use the MinD variant fine-tuned for multi-turn patterns and trained with 100 GRPO steps on the MATH training set.

Table 7 reports a sensitivity sweep on MATH-500, varying the $\mathcal{R}_{\text{unit}}$ weight while keeping all other settings unchanged. A modest penalty improves efficiency with negligible or positive effects on

accuracy; an overly large penalty degrades both. In particular, a small negative weight achieves the best efficiency, whereas a slightly stronger penalty yields the best accuracy, indicating a smooth trade-off rather than a brittle optimum.

Table 7: Sensitivity of MinD to the $\mathcal{R}_{\text{unit}}$ weight on MATH-500. Accuracy (higher is better) and average output token usage (lower is better). All runs use the multi-turn pattern fine-tuned model with 100-step GRPO on MATH.

| Weight for $\mathcal{R}_{\text{unit}}$ | Accuracy | Token |
|---|---|---|
| 0 | 80.0 | 3258.0 |
| −0.3 | 82.0 | **3171.2** |
| −0.5 | **83.6** | 3325.1 |
| −1.0 | 80.4 | 3498.1 |

**Accuracy–Efficiency Balance under Compact Reasoning**    As shown in Table 3, MinD-1.5B delivers substantial efficiency on MATH-500—about 68% fewer output tokens (1719 vs. 5389)—while maintaining competitive accuracy (82.8% vs. 85.4% for the original LRM). The small gap mainly reflects the size and composition of the GRPO set (the MATH training set), which skews toward easier items and nudges the model toward very concise reasoning on harder cases. To narrow this gap without sacrificing compactness, we scale GRPO with harder chain-of-thought data. In a preliminary continuation, training MinD-1.5B for one additional GRPO epoch on a small mixed set (50 MATH + 50 DeepScaleR (Luo et al., 2025b), randomly sampled) reached 84.2% with 1804 average tokens, indicating clear headroom from data scaling.

**Early-exit behavior under forced truncation**    To better understand how MinD enables early exit at the unit level, we perform an additional analysis on MATH-500 with MinD-1.5B. For each generated multi-turn trajectory, we *manually truncate* the reasoning at a chosen turn $k$ by detecting the next `<think>` marker and forcing decoding to stop before it, treating the answer from the previous unit as the final output. As shown in Table 8, compared to the original LRM (85.4% accuracy, 5389 tokens) and the full MinD model after GRPO (82.8%, 1719 tokens), forcing exit at turn 1 already reaches 80.4% accuracy with only 1436 tokens, while forcing exit at turns 2–4 yields 82.6–82.8% accuracy with 1623–1710 tokens. This indicates that (i) intermediate units already contain high-quality answers, and (ii) after GRPO the turn distribution is already concentrated (most samples naturally use only 1–2 turns, cf. Figure 5), so additional forced early exits bring limited further gains and only small accuracy differences across "turn 2/3/4" settings.

Table 8: Effect of forced early exit at different turns on MATH-500 with MinD-1.5B.

|  | Accuracy | Tokens |
|---|---|---|
| Original LRM | 85.4 | 5389 |
| MinD | 82.8 | 1719 |
| Forced exit at turn 1 | 80.4 | 1436 |
| Forced exit at turn 2 | 82.6 | 1623 |
| Forced exit at turn 3 | 82.8 | 1689 |
| Forced exit at turn 4 | 82.8 | 1710 |

**Word Frequency Analysis of Thinking Units**    We collect and compare the number of distinct words representing thinking units in DeepSeek-R1-Distill-1.5B, including the Original LRM, Non-Multi-Turn (GRPO applied without explicit multi-turn segmentation) , and MinD. Although these words do not precisely correspond to the number of actual thinking units, they serve as a meaningful proxy and offer indicative insights into their distribution(see Table 9 for details).

Table 9: The frequency of words representing thinking units in outputs generated by Original LRM, Non-Multi-Turn and MinD across MATH-500, AIME24 and AMC23.

| | Wait | Alternatively | double-check | check | verify |
|---|---|---|---|---|---|
| **MATH-500** | | | | | |
| Original LRM | 13993 | 2206 | 368 | 1272 | **124** |
| Non-Multi-Turn | 1822 | 333 | 41 | **347** | 193 |
| MinD | **1651** | **237** | **10** | 434 | 249 |
| **AIME24** | | | | | |
| Original LRM | 3742 | 415 | 20 | 215 | 17 |
| Non-Multi-Turn | 317 | 67 | **0** | 45 | 19 |
| MinD | **211** | **45** | **0** | **34** | **8** |
| **AMC23** | | | | | |
| Original LRM | 2302 | 385 | 35 | 205 | 45 |
| Non-Multi-Turn | 246 | 38 | 3 | **42** | **17** |
| MinD | **215** | **30** | **0** | 50 | 22 |

# B    EXPERIMENT SETTING

We use DeepSeek-R1-Distill-Qwen-1.5B/7B (DeepSeek-AI, 2025) as base reasoning models. For the initial supervised fine-tuning (SFT) phase, full-parameter tuning is employed over 2 epochs, with a learning rate of 5e-5, a batch size of 4, and fp16 precision. During the GRPO phase, training is performed for 1 epoch, where the actor learning rate is set to 1e-6. The model generates 10 rollout completions per sample via a vLLM-based rollout backend. All GRPO training is conducted on the MATH (Lightman et al., 2023) training set.

For the evaluation in Table 3, we utilised Open-R1 (Face, 2025) as the core framework. All decoding hyper-parameters are held constant across tasks: maximum response length of 32,768 tokens, temperature = 0.6, and top-p = 0.95. For the larger benchmarks (namely MATH-500, GPQA-Diamond and LiveCodeBench) we report metrics averaged over four independent runs; for the smaller datasets (AIME24 and AMC23), owing to the reduced sample size, we increased the number of independent trials to sixteen to enhance statistical reliability. When publicly available checkpoints existed (e.g., ShorterBetter, AdaptThink, ThinkPrune) we applied the same decoding settings; for other baselines we adhered to the values reported in their original publications.

# C    PROMPTING FOR MIND

In this section, we present the complete prompt formats used in the MinD process (see Figure 3 for details).

---

**Q&A Template**

```
{Question}
Please reason step by step, and put your final answer within
\\boxed{}.
```

---

**Decomposing into Thinking Units**

```
You will be provided with a math problem and a solution
generated by a reasoning model.  The model's response may
contain multiple Reasoning Rounds.  One Reasoning Round
is a part of the full model generation and is defined as
a complete reasoning process or verification process that
explicitly contains the final answer.  Your task is to
carefully analyze the response and segment it into individual
Reasoning Rounds.  Specifically, insert "[split]" between
every two consecutive Reasoning Rounds.
--
Problem:  {question}
Solution:  {prediction}
--
Please give the solution with "[split]" tags without any
redundant words.
```

## D    STATEMENT ON THE USE OF LLM ASSISTANCE

Consistent with community guidelines on responsible use of large language models (LLMs), we disclose that LLM tools were used only to assist with language editing (grammar, wording, and minor style) of this manuscript. All ideas, claims, methods, experiments, analyses, figures, and tables were conceived, implemented, and verified by the authors. The authors reviewed and edited all LLM-suggested text for accuracy and clarity; no passages were accepted without human verification. LLMs were not used to generate data, code, results, reviews, or citations, and no confidential or proprietary information was provided to LLM services.