# OpenReview forum: "Done Is Better than Perfect: Unlocking Efficient Reasoning by Structured Multi-Turn Decomposition"
_ICLR.cc/2026/Conference — Submitted to ICLR 2026_

### Official Review · Reviewer_quxH · 2025-10-17

**Soundness:** 2
**Presentation:** 2
**Contribution:** 2
**Rating:** 4
**Confidence:** 4

**Summary:**

The excessively lengthy Chain-of-Thought (CoT) of Large Reasoning Models (LRMs) substantially increases computational costs and latency. To mitigate this, the paper introduces Multi-Turn Decomposition (MinD), a method designed to improve the reasoning efficiency of LRMs by restructuring the traditional "think-then-answer" CoT process into a structured multi-turn format. Following a supervised fine-tuning (SFT) then reinforcement learning (RL) paradigm, LRMs are trained to provide an answer after every thinking turn and keep every thinking turn brief.

**Key Contributions:**

1. A structured multi-turn reasoning paradigm, where long CoT sequences are decomposed into explicit, turn-based interactions.

2. Experiments: significantly reduces the number of reasoning turns, token usage and time-to-first-token (TTFT), while maintaining competitive accuracy.

**Strengths:**

1. The idea of decomposing long CoT into multiple intermediate thinking turns and providing intermediate answers is novel.
2. The organization of this paper is logical, and the background work and methodology are easy to understand.
3. This paper focuses on reducing the output length of reasoning models, which is an important issue in the field.

**Weaknesses:**

1. In Section 3.3, the proposed method is not introduced clearly.

   - The "correct final answer" in Figure 3 and the conventional accuracy reward $\mathcal{R}_{\text{accuracy}}$. Section 3.2  suggests that SFT data is filtered by the "correct final answer," implying the use of complete reasoning chains. Given this, how does the model develop the ability described in Section 3.3 to actively "choose to continue or abort" its reasoning? The observation in 3.3 that the model "tends to generate even more output tokens" further highlights this issue.

   - Unit Compactness Reward $\mathcal{R}_{\text{unit}}$. The criterion for determining "**contains multiple exploratory trajectories**" requires clarification. In Section 3.2, trajectory splitting is performed by an LLM. Is a similar approach used in RL training process? A more detailed explanation would help the community benefit more from this work. Furthermore, if an LLM is invoked for this judgment *during* training, the associated  API costs would be substantial. I recommend that these potential costs be explicitly stated to underscore the practical considerations of the proposed method.

2. Experiments. The extremely small sizes of the AIME24 (N=30) and AMC23 (N=40) datasets raise concerns regarding the statistical significance of the results reported in Tables 3 and 4. Conducting only a single run makes it difficult to draw reliable conclusions. Notably, the baseline methods DEER [0] and ThinkPrune [1] in Table 3 have already set a precedent by reporting averaged results over multiple runs.
3. About time-to-first-token (TTFT). Since the method does not appear to modify the model’s architecture or its initial processing stage, the connection between the proposed approach and the improvement in Time-To-First-Token (TTFT) is unclear to me.

References:

[0] Yang, Chenxu, et al. "Dynamic Early Exit in Reasoning Models." *arXiv preprint arXiv:2504.15895* (2025).

[1] Hou, Bairu, et al. "Thinkprune: Pruning long chain-of-thought of llms via reinforcement learning." *arXiv preprint arXiv:2504.01296* (2025).

**Questions:**

Please see weaknesses for the questions. Besides, given that the SFT model already performs explicit reasoning while its outputs are not shorter, the observed length reduction appears to stem from $\mathcal{R}_{\text{unit}}$.

However, the result in Table 5—where length decreases even with a zero $\mathcal{R}_{\text{unit}}$ weight—is highly unexpected and seems to contradict this intuition. The empirical conclusions in Section 4.3 and discussion on "explicit reward term regarding the number of turns" could be strengthened by a more in-depth discussion of this issue. For example, presenting the learning curves (metrics vs. RL steps) for the experiments in Table 5 might help illustrate how the model’s behavior evolves and offer clearer insights into the underlying factors driving length compression.

---

> ### Author Response · Authors · 2025-11-22
>
> Thank you for your careful and constructive review. Below we respond point-by-point.
>
> ---
>
> ## Main Comments
>
> > **W1-1: Clarity of Section 3.3 and the role of “correct final answer” v.s. the ability to continue/abort reasoning.**
>
> **A:** We clarify the intended meaning and have updated the wording in **Sec. 3.3 (L260)** accordingly.
>
> - SFT by itself does **not** make the model automatically learn when to stop early; it mainly reshapes the format from single-turn *think-then-answer* to **multi-turn reasoning units with intermediate answers**.
> - The phrase “choose to continue or abort” in the original text was meant to indicate that, **once we have a multi-turn pattern**, an external controller (or user) can **manually control** where to stop. Concretely, we can:
>   - **Manually truncate** the reasoning at a chosen turn \(k\) by detecting the next `<think>` marker and forcing decoding to stop before it, treating the answer from the previous unit as the final output.
>   - Conversely, if the model terminates early but we want to explore more reasoning, we can append another `<think>` to prompt it to continue.
>
> Thus, the multi-turn representation produced by SFT **enables controllability** at the unit level (we can externally decide to continue or abort at turn \(k\)), but **does not itself reduce** the number of units: as we observe, the SFT model often generates **even longer** outputs than the original LRM when run without control. This is why we then introduce **GRPO** in Sec. 3.3: RL is used to **optimize for efficient reasoning traces** (fewer units / tokens) while preserving correctness.
>
> > **W1-2: Unit Compactness Reward — definition of “multiple exploratory trajectories” and whether an LLM is used during RL (cost concerns).**
>
> **A:** We clarify that, as already described in Sec. 4.3 (see lines 469-479), **no external LLM is used during RL training** for the unit compactness reward. During GRPO, we detect units that “contain multiple exploratory trajectories” via a **simple pattern-based heuristic** on the generated text. It incurs **negligible additional computational or monetary cost** beyond standard GRPO.
>
> > **W2: Small sizes of AIME24 (N=30) and AMC23 (N=40) and single-run results.**
>
> **A:** We agree that the small test sizes limit statistical significance. In the **revised version**, we report results on AIME24 and AMC23 **averaged over 16 random seeds** (see Table 3).
>
> > **W3: About time-to-first-token (TTFT) and its connection to MinD.**
>
> **A:** We have clarified our definition of TTFT in Sec. 4.1 (lines 363-364). In our setting:
>
> - TTFT refers to the latency until the **first answer token** that the user observes, i.e., the first token in the **answer segment of the first turn**, rather than the first thinking token.
>
> > **Q1: SFT already uses explicit reasoning but is not shorter, so length reduction seems to stem from R-unit. However, Table 5 shows length decreases even with zero R-unit weight, which is surprising.**
>
> **A:** The key clarification is:
>
> - The primary driver of **length reduction** is the **inherent behavior of GRPO**: when optimizing for correctness, GRPO tends to favor **shorter correct completions** over longer ones with similar reward (as we discuss in the revised Sec. 3.3 L276-284). This effect exists **even when R_unit is set to zero**, which explains the token reduction observed in **Table 4 (Non-Multi-Turn)**.
> - **R_unit** serves a **different purpose**: it regularizes the structure of units to **avoid collapse** back to a single monolithic *think-then-answer* chain. In other words, R_unit helps preserve the intended **multi-turn, unit-level organization** of reasoning rather than being the main source of length compression.

---

> > ### Comment · Reviewer_quxH · 2025-11-27
> >
> > Thank you for the response.
> >
> > While your response has indeed clarified some content of the article, it has consequently revealed more significant issues. To more precisely articulate my level of concern regarding each issue, I will explicitly label them as either 'minor' or 'major'.
> >
> > 1.	“Manual control” : As an additional feature of the method, I find it difficult to position this as a methodological contribution, given that many existing approaches (e.g., S1 [0], Qwen3 [1], and your baseline DEER) have already explored early-stopping. Indeed, manual control can be achieved simply by appending specific tokens that guide the model toward the final answer (e.g., S1's "Final Answer", Qwen3’s “Considering the limited time by the user, I have to give the solution based on the thinking directly now.\n</think>.\n\n”). Crucially, these methods support arbitrary-position stopping, whereas your proposed feature only permits termination after unit generation. Moreover, these methods can serve as plug-in solutions compatible with other approaches that reduce output length, as they solely rely on the model’s inherent capabilities. -- minor
> >
> > 2.	The Unit Compactness Reward constitutes a core component of the methodology and should be introduced in the Methods section, rather than being deferred to post-experimental discussion. I recommend consolidating all methodologically relevant content within the Methods section. -- minor
> >
> > 3.	Experiments: Results of the 7B model on AMC23 in Table 3 are unchanged. Results in Table 4 are unchanged. Additionally, I noticed that the baseline data is quite messy. For methods with open-source weights available (ShorterBetter, AdaptThink), I found that you directly cited the experimental results from the original papers, which resulted in only two sets of results being presented. As for DEER, your results differ from those in the original paper, and I couldn't find the experimental results for the 1.B model in the original paper. -- major
> >
> > 4.	Same as 3. Weakness 1 of Reviewer LPTN’s review. While you stated that details are provided in Appendix B, I found it only contains training configurations. Critical experimental settings (e.g., the number of repetitions and how baseline data was collected) remain unclear. -- major
> >
> > 5.	TTFT: Same as 1. Now I understand the meaning of TTFT, but I still don't see how it demonstrates the advantage of your method. This metric could be easily improved by simply inserting some response forcing tokens (See 1). -- minor
> >
> > 6.	Role of SFT and GRPO: SFT and the reward function of RL is both designed to maintain the output format. If the length reduction primarily stems from GRPO's inherent characteristics, then the paper's core innovation (the decomposition of multi-round reasoning) appears to merely provide a formatting-based cold start. -- major
> >
> > 7.	Insight and the actual role of GRPO: If your understanding of GRPO's properties comes from Dr.GRPO (which I notice you've already cited), then a crucial ablation study would be to remove the length-bias part in Dr.GRPO and observe the effects.  Besides, as the accuracy drops, the ratio of the correct completions is reduced. This is in contradiction to “for correct completions, favor shorter ones over longer ones”. Because the model actually produces less correct completions (and more wrong ones, and for wrong completions longer ones are preferred.). – minor. Given time left for discussion, requesting this additional experiment may be too demanding. Therefore, I recommend you prioritize addressing other major issues first. And I mark this point as minor.
> >
> > References:
> >
> > [0] Muennighoff, Niklas, et al. "s1: Simple test-time scaling." EMNLP’25
> >
> > [1] Yang, An, et al. "Qwen3 technical report." arXiv preprint arXiv:2505.09388 (2025).

---

> > > ### Author Response · Authors · 2025-11-29
> > >
> > > Thank you for your constructive feedback. We address each comment below.
> > >
> > > ---
> > >
> > > ## Main Comments
> > >
> > > > **Q1: “Manual control” and relation to prior early-stopping methods (S1, Qwen3, DEER).**
> > >
> > > **A:** Early stopping and thinking-budget control have been widely explored (e.g., S1, Qwen3, DEER), and we do not claim novelty here. Our main contribution is the **unit-level decomposition** of long CoT traces plus **GRPO training** that reduces the number of units while preserving accuracy. The multi-turn SFT stage only exposes unit boundaries, so any manual stopping at unit ends is merely a practical by-product.
> > >
> > > > **Q2: The Unit Compactness Reward should be introduced in the Methods section.**
> > >
> > > **A:** We agree and have updated the paper accordingly. The Unit Compactness Reward is now defined in **Sec. 3.3**, while **Sec. 4.3** focuses on its empirical effects.
> > >
> > > > **Q3–Q4: Baseline results (ShorterBetter, AdaptThink, DEER), unchanged entries in Tables 3–4, and missing details on repetitions / data collection.**
> > >
> > > **A:** We have revised the experimental section and **Appendix B** to make the evaluation protocol and baseline provenance explicit:
> > >
> > > - **Repetitions.** For larger benchmarks (MATH-500, GPQA-Diamond, LiveCodeBench), all **Table 3** results are now the **average over 4 runs**. For smaller datasets (AIME24, AMC23), we use **16 runs**. These settings apply to MinD and all re-evaluated baselines and are stated in Appendix B.
> > >
> > > - **ShorterBetter / AdaptThink and other baselines.** When public checkpoints exist (e.g., **ShorterBetter**, **AdaptThink**, **ThinkPrune**), we evaluate them ourselves on all benchmarks using a unified decoding setup (same max length, temperature, top-p as MinD). For methods without released checkpoints (including **DEER**), we use the numbers reported in their papers; Appendix B now clarifies for each method whether the value is from our re-run or from the original paper. In particular, the **1.5B** results for DEER are taken from an earlier arXiv version (v1), where 1.5B models were explicitly reported.
> > >
> > > > **Q5: TTFT can be easily improved by inserting response-forcing tokens; how does TTFT demonstrate the advantage of your method?**
> > >
> > > **A:** We use TTFT as part of the **time–accuracy trade-off**, not as a standalone novelty. Under the **same decoding setup** (no extra “stop-thinking” tokens or special prompts), MinD achieves **lower TTFT and token usage at similar or better accuracy** than the Original LRM and the early-exit baseline DEER (Table 3). Thus, the gain comes from the **learned behavior of MinD** (unit-level decomposition + GRPO), not from test-time tricks. We agree that forcing tokens could further reduce TTFT for any model; such manual controls are **orthogonal** and could in principle be combined with MinD.
> > >
> > > > **Q6: Role of SFT and GRPO; is multi-turn decomposition just a formatting-based cold start if length reduction mainly comes from GRPO?**
> > >
> > > **A:** SFT (multi-turn decomposition) and GRPO are **complementary**. SFT does not shorten CoT; it restructures it into **semantically coherent units with intermediate answers**. GRPO then operates on this structure: instead of compressing tokens within one long trace, it learns to **reduce the number of units** while keeping each unit’s internal reasoning intact.
> > >
> > > This is more than a formatting cold start. In the **Non-Multi-Turn** ablation, where GRPO is applied directly to standard CoT, we obtain similar in-distribution performance on MATH-500 but **worse out-of-distribution accuracy and smaller token savings** than MinD. This indicates that the multi-turn formulation genuinely changes the optimization landscape and is needed for the efficiency–accuracy gains we report, while still keeping the method simple and easy to integrate into existing LRM pipelines.
> > >
> > > > **Q7: Insight and the actual role of GRPO; suggestion to remove the length-bias term as an ablation.**
> > >
> > > **A:** In our setting, the relevant comparison for GRPO is the **SFT-only** model, not the original LRM. As shown in Table 4, GRPO **does not reduce accuracy relative to SFT-only** (and sometimes improves it), while clearly shortening the outputs, so the contradiction you describe does not occur in our experiments.
> > >
> > > We do not modify the GRPO objective and do not aim to re-study its intrinsic length bias, which has already been analyzed in prior work (e.g., Dr.GRPO). We instead **leverage this known behavior** under a different formulation: after multi-turn decomposition, GRPO operates in the unit-level space. Consistent with these analyses, our Non-Multi-Turn ablation (GRPO on standard CoT without multi-turn segmentation) also shortens outputs, indicating that length reduction is a general property of GRPO rather than specific to our decomposition. An additional ablation that removes the length-bias term inside GRPO would be interesting but is orthogonal to our contribution and not strictly necessary for this paper.

---

### Official Review · Reviewer_LPTN · 2025-10-20

**Soundness:** 2
**Presentation:** 4
**Contribution:** 3
**Rating:** 4
**Confidence:** 4

**Summary:**

The authors propose the Multi-Turn Decomposition framework to address the issues of excessively lengthy outputs, high time to first token, and the inability to explicitly manage thinking units in the Chain-of-Thought reasoning of Large Reasoning Models. This framework reconstructs Chain-of-Thought through a two-stage process of Supervised Fine-Tuning and Reinforcement Learning: in the Supervised Fine-Tuning stage, another Large Language Model is used to convert single-segment Chain-of-Thought into an explicit multi-turn format of "thinking units—intermediate answers"; in the Reinforcement Learning stage, a tripartite reward signal based on GRPO is employed to suppress redundancy, solving the problem of increased token consumption after Supervised Fine-Tuning. Trained on the MATH dataset, Multi-Turn Decomposition reduces both output token usage and time to first token by approximately 70% while maintaining competitive performance across multiple benchmark tests, filling the gap in the regulation of traditional Chain-of-Thought and providing a new path for efficient reasoning in Large Reasoning Models.

**Strengths:**

The authors propose a concise method to compress the reasoning process of Large Reasoning Models, which effectively reduces the number of tokens consumed during inference. Additionally, the paper is clearly written, allowing readers to easily follow the content and grasp the key ideas.

**Weaknesses:**

1. Insufficient reproducibility: The paper fails to provide detailed experimental settings for both the proposed method and baselines, such as specific hyperparameter configurations and whether repeated experiments were performed. This lack of key information hinders other researchers from replicating the study’s results and verifying its conclusions.
2. Limited diversity of baselines: Most baselines in the experiments fall into the category of methods that control early stopping via token budget constraints. There is no comparison with other representative approaches for token optimization, including those based on length penalty rewards, reasoning path pruning, and reasoning token simplification. This narrow baseline scope restricts a comprehensive assessment of the proposed method’s competitiveness in reducing inference tokens.

**Questions:**

1. The paper would benefit from comparing the proposed method with a wider range of reasoning compression approaches, alongside a detailed discussion of the advantages and drawbacks of each category. Such a comparison would more prominently highlight the unique strengths of MinD in token reduction and inference efficiency.
2. If the method were applied to other model families (e.g., QwQ), would similar or different findings emerge?
3. What impact would providing positive $R_{unit}$ values during Compliance have on the model’s performance?

---

> ### Author Response · Authors · 2025-11-22
>
> Thank you for your careful and constructive review. Below we respond point-by-point.
>
> ---
>
> ## Main Comments
>
> > **W1: Insufficient reproducibility (missing experimental settings, hyperparameters, and information about repeated runs).**
>
> **A:** We appreciate this comment and agree that more details are needed for reproducibility. These details are included in Appendix B.
>
> > **W2 & Q1: Limited diversity of baselines; need comparison with a wider range of reasoning compression approaches (length penalty, reasoning path pruning, token simplification) and a clearer discussion of pros/cons.**
>
> **A:** Thank you for this suggestion. In the **revised version**, we reorganize and expand our discussion of reasoning compression methods into four categories:
>
> 1. **RL-based length optimization / pruning** (e.g., ThinkPrune, ShorterBetter), which add length-aware rewards or clipping inside GRPO to shorten CoT while preserving correctness.
> 2. **Training-free dynamic early-exit decoding** (e.g., DEER, Dynasor), which monitor confidence / consistency signals during generation to decide when to stop.
> 3. **Adaptive thinking-mode selection** (e.g., AdaptThink), which learns when to use long “Think” reasoning vs. short “NoThink” answers.
> 4. **Token-budget–based early stopping** (our original baselines), which control length via explicit token limits at decoding time.
>
> We now explicitly position **MinD** relative to these categories in **Table 3, Table 8 and the Related Work section**.
>
> > **Q2: If the method were applied to other model families (e.g., QwQ), would similar or different findings emerge?**
>
> **A:** Due to computational constraints, we could not evaluate on QwQ specifically, but we **did extend** our experiments to **two additional model families** beyond the original DeepSeek-R1-Distill-Qwen:
>
> - **DeepSeek-R1-Distill-Llama3.1-8B** on MATH-500:
>
>   | Model                    | Accuracy (%) | Tokens   |
>   |--------------------------|-------------:|---------:|
>   | R1-Llama3.1-8B           | 78.0         | 4792.37  |
>   | R1-Llama3.1-8B-SFT       | 75.2         | 5068.48  |
>   | R1-Llama3.1-8B-MinD      | 77.4         | 3107.89  |
>
> - **Qwen3-1.7B** on MATH-500:
>
>   | Model                    | Accuracy (%) | Tokens   |
>   |--------------------------|-------------:|---------:|
>   | Qwen3-1.7B               | 91.0         | 5216.44  |
>   | Qwen3-1.7B-SFT           | 88.6         | 5433.30  |
>   | Qwen3-1.7B-MinD          | 89.2         | 3866.69  |
>
> Across these additional families, MinD **reduces tokens by ~25–35%** while keeping accuracy close to the original LRM, indicating that our findings are **not specific** to a single base model. We include these results in Table 5.
>
> > **Q3: What impact would providing positive R-unit values during Compliance have on the model’s performance?**
>
> **A:** R_unit is introduced as an regularizer to discourage units that contain multiple exploratory trajectories, and to encourage cleaner, one-trajectory-per-unit behavior.
>
> The dominant factor in controlling overall trace length is the **inherent length bias of GRPO**, which prefers shorter correct completions.
>
> Therefore, using a positive R_unit versus setting it to zero makes little practical difference: the model still tends to avoid unnecessarily long trajectories. Multi-turn traces will collapse into a think-then-answer style, where a single thinking unit contains multiple trajectories (as shown in Figure 6, right).

---

> ### Comment · Reviewer_LPTN · 2025-11-26
>
> I appreciate the authors' work during the rebuttal period and the extra experiments they conducted to address my concerns. After reviewing the new results and the revised paper, I have decided to raise my rating.

---

> > ### Author Response · Authors · 2025-11-26
> >
> > Dear Reviewer LPTN,
> >
> > Thank you for considering our rebuttal and for increasing the score. We are grateful for your feedback.
> >
> > Best regards,
> >
> > The Authors

---

### Official Review · Reviewer_ACT4 · 2025-10-30

**Soundness:** 2
**Presentation:** 3
**Contribution:** 2
**Rating:** 4
**Confidence:** 4

**Summary:**

This paper introduces a methodology to reduce overthinking in the reasoning trace of Chain-of-Thought (CoT)-based ‘reason-then-answer’ methodology employed in off-the-shelf large Reasoning Models (LRM). This methodology adapts the LRM from ‘reason-then-answer’ to ‘Mult-Turn-Decomposition’ (MinD) via Supervised Finetuning (SFT), followed by Reinforcement Learning based GRPO, enabling the generated response as a set of independent reasoning traces, each containing a final answer. Hence, this output format enables early exit and reduces latency and token utilization. Through experiments on two mathematical question answering datasets, they empirically verify the reduction in token usage during the generation of reasoning traces to arrive at the final answer with reduced output latency.

**Strengths:**

- The research problem under study, i.e., ‘overthinking’ in reasoning generation to arrive at the final answer, is very prominent in LRMs, resulting in frequent wrong generation and latency overload compared to non-LRMs. This work introduces a competitive methodology for addressing this issue.
- The proposed methodology reformulates the CoT mechanism in LRMs into multi-turn decomposition, where each turn provides independent reasoning with a candidate final answer, allowing early exit and reduced token utilization (e.g., a 68% reduction on MATH-500 for 1.5B) and redundancy in the reasoning chain (Fig. 2, left).
- The proposed methodology also enables reduced total latency as the computation time between processing the input prompt and final answer generation (Fig. 4).

**Weaknesses:**

- **Generalization of empirical results beyond the choice of LRM**: The entire experiments are conducted considering only one reasoning model (DeepSeek-R1-Distill-Qwen) with two different model sizes (1.5B and 7B). Though the empirical results regarding reduced token utilization and latency while maintaining model accuracy are clearly demonstrated for the Qwen model, the absence of incorporating other reasoning-based models (eg, DeepSeek-R1-Distill-Llama-8B) limits its generalization and reliability across diverse mode families of LRMs.
- **Generalization across diverse knowledge-intensive tasks**: The experiments are conducted on a mathematical question answering dataset. However, it would be interesting to analyze the reduction in reasoning trace using MinD on non-mathematical datasets such as commonsense reasoning Q&A datasets with varying reasoning depths (eg, TruthfulQA, OpenBookQA).
- **Proper definition of thinking units for segmentation of CoT reasoning**: The segmentation of CoT reasoning into a set of independent reasoning units is done using GPT-4o. The quality of this segmentation is crucial for SFT and GRPO-based training. However, there is no error analysis on CoT segmentation quality (through annotation) and the proposed MinD performance.
- **Lack of proper description of the mechanism of early exit**: The proposed mechanism converts the CoT-based ‘reason-then-answer’ to a sequence of independent thinking uint delimited by <think></think> (lines 254-255), each containing a candidate final answer. However, there is no clear discussion on the early exit mechanism.

**Questions:**

Please see the weaknesses above.

---

> ### Author Response · Authors · 2025-11-22
>
> We appreciate your detailed and insightful review. Our responses to your points are given below.
>
> ---
>
> ## Main Comments
>
> > **W1: Generalization of empirical results beyond the choice of LRM (only DeepSeek-R1-Distill-Qwen).**
>
> **A:** We additionally evaluate **DeepSeek-R1-Distill-Llama3.1-8B** and **Qwen3-1.7B** on **MATH-500** using the same MinD pipeline:
>
> | Model                    | Accuracy (%) | Tokens   |
> |--------------------------|-------------:|---------:|
> | R1-Llama3.1-8B           | 78.0         | 4792.37  |
> | R1-Llama3.1-8B-SFT       | 75.2         | 5068.48  |
> | R1-Llama3.1-8B-MinD      | 77.4         | 3107.89  |
>
> | Model                    | Accuracy (%) | Tokens   |
> |--------------------------|-------------:|---------:|
> | Qwen3-1.7B               | 91.0         | 5216.44  |
> | Qwen3-1.7B-SFT           | 88.6         | 5433.30  |
> | Qwen3-1.7B-MinD          | 89.2         | 3866.69  |
>
> Across these additional families, MinD consistently **reduces tokens (≈25–35%)** while maintaining accuracy close to (or slightly below) the original LRM. This supports that MinD is **not specific to R1-Distill-Qwen** but is a model-agnostic training recipe. The full results and discussion are included in Appendix A.
>
> > **W2: Generalization across diverse knowledge-intensive tasks.**
>
> **A:** We have added an experiment on **OpenBookQA**:
>
> | Model           | Accuracy (%) | Tokens   |
> |-----------------|-------------:|---------:|
> | R1-1.5B         | 27.4         | 4986.47  |
> | R1-1.5B-SFT     | 31.0         | 5433.30  |
> | R1-1.5B-MinD    | 34.8         | 3840.93  |
>
> On this dataset, MinD **improves accuracy** over the original LRM while also **reducing token usage**, indicating that unit-level multi-turn reasoning is beneficial beyond mathematical domains. We will report these results in Appendix A.
>
> > **W3: Proper definition of thinking units and segmentation quality of CoT reasoning.**
>
> **A:** We now provide a **quantitative error analysis** of GPT-4o-based unit segmentation:
>
> - On **150 randomly sampled GSM8K training examples**,
>   - **127/150 (84.67%)** segmentations fully satisfy our unit criteria (each unit is a coherent sub-step ending with a candidate answer).
>   - **138/150 (92.00%)** have **≤2 minor boundary errors** that do not change the core reasoning semantics.
>
> > **W4: Lack of proper description of the mechanism of early exit.**
>
> **A:** We conduct an additional analysis on MATH-500 using the MinD-1.5B. We **manually truncate** the generated trajectory at different turns by detecting the next `<think>` marker and forcing the model to stop before it, treating the answer from the previous unit as the final output. The resulting accuracy and average output length are:
>
> | Setting               | Accuracy (%) | Avg. tokens |
> |-----------------------|-------------:|------------:|
> | Original LRM          | 85.4         | 5389        |
> | MinD (GRPO)           | 82.8         | 1719        |
> | Forced exit at turn 1 | 80.4         | 1436        |
> | Forced exit at turn 2 | 82.6         | 1623        |
> | Forced exit at turn 3 | 82.8         | 1689        |
> | Forced exit at turn 4 | 82.8         | 1710        |
>
> Two observations:
>
> 1. **Intermediate units already contain high-quality answers:** even when we force exit at turn 1 or 2, accuracy remains close to the full MinD model, while tokens are further reduced.
> 2. **The distribution of turns is already concentrated:** as shown in **Figure 5**, after GRPO most samples naturally use only 1–2 turns, so forcing additional early exits has limited further impact, which explains the small accuracy differences across “turn 2/3/4” settings.
>
> This analysis has been added to Appendix A in the revised paper.

---

### Official Review · Reviewer_nS19 · 2025-10-31

**Soundness:** 3
**Presentation:** 2
**Contribution:** 2
**Rating:** 6
**Confidence:** 4

**Summary:**

This paper aims to address the issue of excessive token usage of LRMs. The authors propose the MinD (Multi-Turn Decomposition) method. Specifically, they first conduct SFT to segment and restructure CoT data into a structured, multi-turn format, where each turn is a "reasoning unit". Then, they leverage the GRPO algorithm's "implicit bias" for short sequences to incentivize the model to use fewer reasoning turns to reach the correct final answer. The method achieves good accuracy and reduces the number of tokens in both in-domain and OOD data.

**Strengths:**

1. The paper conducts an exploration of reasoning unit redundancy, which better demonstrates the motivation rather than just directly claiming LRM tokens are redundant.

2. A novel method design that utilizes GRPO's implicit ability for shorter turns rather than incorporating a length penalty directly into the reward function.

**Weaknesses:**

1. Regarding the reasoning units, it is necessary to measure the quality of the obtained unit splits and whether they clearly reflect the LRM's reasoning process.

2. The authors explored too few LRMs with only two models distilled from DeepSeek-R1. This makes it difficult to prove whether the unit segmentation is overly dependent on DeepSeek-R1's text style, and it also needs to be verified if the multi-turn training method is applicable to other LRMs.

3. More baselines could be considered, such as [1][2].

[1] Yi, Jingyang, Jiazheng Wang, and Sida Li. "Shorterbetter: Guiding reasoning models to find optimal inference length for efficient reasoning.".

[2] Zhang, Jiajie, et al. "Adaptthink: Reasoning models can learn when to think, 2025".

**Questions:**

1. Why does MinD achieve much higher accuracy than the Original LRM on AMC23 but not on other datasets?

2. In Table 4, why does SFT-Only lead to a decrease in accuracy? It seems that it merely changes the format and does not require a reduction in reasoning steps.

---

> ### Author Response · Authors · 2025-11-22
>
> Thank you for your careful and constructive review. Below we respond point-by-point.
>
> ---
>
> ## Main Comments
>
> > **W1: Regarding the reasoning units, it is necessary to measure the quality of the obtained unit splits and whether they clearly reflect the LRM's reasoning process.**
>
> **A:** We have now **quantitatively evaluated** our splitting procedure on **150 randomly sampled GSM8K training examples**:
>
> - **127/150 (84.67%)** decompositions fully satisfy our unit criteria (each unit is a coherent sub-step ending with a candidate answer).
> - **138/150 (92.00%)** have at most **two minor boundary errors** that do not change the underlying reasoning semantics.
>
> > **W2: The authors explored too few LRMs with only two models distilled from DeepSeek-R1. This makes it difficult to prove whether the unit segmentation is overly dependent on DeepSeek-R1's text style, and it also needs to be verified if the multi-turn training method is applicable to other LRMs.**
>
> **A:** To address this, we additionally evaluate **Qwen3-1.7B**, which is **not** distilled from DeepSeek-R1, using the same segmentation and MinD training pipeline on **MATH-500**:
>
> | Model                | Accuracy (%) | Tokens   |
> |----------------------|-------------:|---------:|
> | Qwen3-1.7B           | 91.0         | 5216.44  |
> | Qwen3-1.7B-SFT       | 88.6         | 5433.30  |
> | Qwen3-1.7B-MinD      | 89.2         | 3866.69  |
>
> MinD still yields a **substantial token reduction (~26%)** with accuracy close to the original model, indicating that (1) our multi-turn training is applicable beyond DeepSeek-style models, and (2) the segmentation is not overly dependent on DeepSeek-R1’s text style. The full results and discussion have been added to Appendix A.
>
> > **W3: More baselines could be considered, such as [1] ShorterBetter and [2] AdaptThink.**
>
> **A:** We have incorporated **ShorterBetter** and **AdaptThink** into our discussion of related work, highlighting that these methods adapt the **token-level** reasoning length/mode at inference time, whereas MinD operates at the **unit level** via multi-turn decomposition and unit pruning. We also provide a more detailed comparison and additional analysis in **Table 3**.
>
> Besides, both [1] and [2] are **contemporaneous** with our submission. According to the ICLR policy on contemporaneous work (“We consider papers contemporaneous if they are published within the last four months ...”), such papers are not strictly required to be included as experimental baselines.
>
> > **Q1: Why does MinD achieve much higher accuracy than the Original LRM on AMC23 but not on other datasets?**
>
> **A:** AMC23 is a challenging benchmark that contains **longer, competition-style problems** with multiple intertwined sub-steps. In our experiments, we observe that:
>
> - The **Original LRM** tends to produce **very long and sometimes meandering CoT traces** on AMC23, occasionally “losing the thread” of the solution despite having strong local reasoning ability.
> - MinD’s **unit-level decomposition + RL** encourages the model to:
>   1. Organize reasoning into **shorter, self-contained units with intermediate answers**, and
>   2. **Prune redundant or unhelpful units**, thus reducing overthinking.
>
> Our hypothesis is that on AMC23, this structure acts as a **form of regularization**: it encourages the model to solve the problem through clearer sub-goals and avoid drifting into overly long, error-prone derivations. This leads to a **larger accuracy gain** relative to the Original LRM.
>
> > **Q2: In Table 4, why does SFT-Only lead to a decrease in accuracy? It seems that it merely changes the format and does not require a reduction in reasoning steps.**
>
> **A:** Importantly, **SFT-Only is not just a cosmetic format change**. The model is **retrained** to follow a new multi-turn pattern with intermediate answers, which alters:
>
> - how information is distributed across tokens,
> - where the final answer appears, and
> - the internal notion of what a “solution trajectory” looks like.
>
> Given our small SFT data, this shift can partially disrupt the strong reasoning behavior learned during pretraining/distillation, without any explicit optimization for end-task reward, leading to an accuracy drop.
>
> MinD adds an RL stage on top of this multi-turn format, directly optimizing for **correctness and efficiency**, which recovers (and often improves upon) the original accuracy while reducing tokens.

---

### Author Response · Authors · 2025-11-27

Dear Reviewers and ACs,

Thank you very much for your constructive feedback. We have carefully addressed all comments in our rebuttal and revised the paper accordingly, with all changes marked in **red**.

We believe these revisions have improved the clarity and quality of the paper. We would greatly appreciate it if the ACs could kindly invite the reviewers to consider our rebuttal and the revised manuscript and update their evaluations if appropriate.

Best regards,

The authors

---

### Meta-Review · Area_Chair_8bhY · 2026-01-12

**Summary:**

The paper aims to improve the reasoning efficiency of LRMs by (1) decomposing/segmenting CoT into multiple turns of trials, i.e., thinking units, by GPT4o; (2) enforcing the model to generate answers after each unit, turning each unit into "think and then answer" format; (3) finetuning the model (SFT) to produce the new formated sequences of thinking units; and (3) applying GRPO with rewards that keep enforcing the format. Due to the inductive bias of GRPO on distributing rewards to tokens, the GRPO stage favors shorter reasoning sequences and thus reduces the reasoning turns. Experiments on existing LRMs show that the approach can effective reduce the number of turns and token counts without compromising the performance.

**Reviewer Concerns:**

- Experiments are limited to one model family.
- Experiments are limited to reasoning benchmarks (some are very small) without knowledge-intensive tasks.
- Unclear roles of SFT and GRPO, and how they contribute to the improvement of reasoning efficiency.
- Unclear utility of the unit compactness reward.
- Quality of thinking unit segmentation relies on GPT4-o, whose performance is not evaluated.
- Lack more comprehensive comparisons with baselines.
- Some evaluation metrics, such as TTFT, are not properly explained.

**Reviewer Scores:**

- The initial ratings from reviewers are 4, 4, 4, 6, all with a confidence of 4.
- The authors' response promisingly addressed the main concerns about experiments and provided some clarifications of key components. Based on this, one reviewer with an original rating of 4 decided to raise the rating, making the paper a typical borderline paper.
- Another reviewer with a rating of 4 is also involved in the after-rebuttal discussion, but the discussion did not convince the reviewer to raise the rating.
- I carefully read all the comments and discussions. Despite its gain in reasoning efficiency, it is not entirely clear how such a gain comes from the newly formatted reasoning sequence (it is clear that the GRPO's inductive bias encourages shorter reasoning). One key question that I cannot find an answer to is: if some rollouts in GRPO contain correct answers in earlier turns, will the reward capture it and encourage early-stopping? It seems that the successful reward only depends on the correctness of the final answer. If so, what is the purpose of training the model (via SFT and then GRPO with formatting reward) to follow the new think-and-answer format?
- The technical novelty of this paper is limited to some extent: (1) the GRPO's inductive bias on shorter responses is not the contribution of this paper; (2) the segmentation of reasoning traces by thinking patterns/keywords is not novel and common in previous works; (3) the main remaining novelty is the new formatting, but it is hard to build a connection between it and the efficiency gain.
- The clarity of this paper has a large room for improvement, especially the GRPO part. For example, did you still use single-turn GRPO or multi-turn RL (given the multi-turn nature of reasoning traces)? How did you define each rollout?

---

### Decision · Program_Chairs · 2026-01-26

Reject